# Experiences of public-private contracting for caesarean delivery in rural district public hospitals: A qualitative interview study

**Tanya Doherty**[1,2,3]*, **Sue Fawcus**[4], **Emmanuelle Daviaud**[1], **Yvandi Bartmann**[1], **Geetesh Solanki**[1,5]

**1** Health Systems Research Unit, South African Medical Research Council, Cape Town, South Africa, **2** Department of Paediatrics and Child Health, University of Cape Town, Cape Town, South Africa, **3** School of Public Health, University of the Western Cape, Cape Town, South Africa, **4** Department of Obstetrics and Gynaecology, University of Cape Town, Cape Town, South Africa, **5** Health Economics Unit, University of Cape Town, Cape Town, South Africa

* tanya.doherty@mrc.ac.za

**Data Availability Statement:** All relevant data are within the paper.

**Funding:** This work was supported by the Bill and Melinda Gates Foundation with a grant to the South

## Abstract

Governments in sub-Saharan Africa are exploring public-private-engagements for the delivery of health services. While there is existing empirical literature on public-private-engagements in high-income countries, we know much less about their operation in low and middle-income countries. Obstetric services are a priority area where the private sector can make an important contribution in terms of skilled providers. The objective of this study was to describe the experiences of managers and generalist medical officers, of private general practitioner (GP) contracting for caesarean deliveries in five rural district hospitals in the Western Cape, South Africa. A regional hospital was also included to explore perceptions of public-private contracting needs amongst obstetric specialists. Between April 2021 and March 2022, we conducted 26 semi-structured interviews with district managers (n = 4), public sector medical officers (n = 8), an obstetrician in a regional hospital (1), a regional hospital manager (1) and private GPs (n = 12) with public service contracts. Thematic content analysis using an inductive, iterative approach was applied. Interviews with medical officers and managers revealed justifications for entering into these partnerships, including retention of medical practitioners with anaesthetic and surgical skills and economic considerations in staffing small rural hospitals. The arrangements held benefits for the public sector in terms of bringing in required skills and having after hours cover; and for the contracted private GPs who could supplement their income, maintain their surgical and anaesthetic skills and keep up to date with clinical protocols from visiting specialists. The arrangements held benefits for both the public sector and the contracted private GPs and were deemed to be an example of how national health insurance could be operationalised for rural contexts. Perspectives of a specialist and manager from a regional hospital provided insight into the need for different public-private solutions for this level of care in which contracting out of elective obstetric services should be considered. The sustainability of any GP contracting arrangement, such as described in this paper, will require ensuring that medical education programmes include basic surgical and anaesthetic skills training so that GPs opening

African Medical Research Council (Grant
Agreement ID INV-023276). The funders had no
role in study design, data collection and analysis,
decision to publish, or preparation of the
manuscript.

practice in rural areas have the required skills to provide these services for district hospitals where needed.

## Introduction

The health-related Sustainable Development Goal 3 includes the target of universal health coverage (UHC). UHC aims to ensure access to needed services of sufficient quality for everyone without suffering financial hardship [1]. Health financing reforms are one means towards reaching this aim, including through public-private engagement (PPEs). PPE's are defined as 'the deliberate, systematic collaboration of the government and the private health sector according to national health priorities, beyond individual interventions and programmes with the aim of harnessing private sector resources to further public health goals [2]. Many governments in sub-Saharan Africa are seeking to establish PPEs for health care infrastructure and delivery of health services. PPEs can take a variety of forms and while there is existing empirical literature on PPEs in high-income countries, we know less about their operation in low and middle-income countries (LMICs) [3, 4].

Health services in South Africa are provided through both a private and a public sector. The public health sector serves more than 80% of the 48 million South Africans without private health insurance. However, 10% of the uninsured population use private sector services on an out-of-pocket basis, and 24% of the insured population use public sector services [5]. The South African government is preparing to implement a national health insurance scheme [6] as a mechanism towards provision of UHC. Under national health insurance, public resources will be used to strategically purchase health services for the entire population from both the public and private sectors [7].

Low and middle income countries including South Africa face considerable challenges in ensuring access to safe and appropriate obstetric care for rural districts especially access to caesarian delivery [8]. Obstetric services are a priority area where the private sector can make an important contribution in terms of skilled providers due to the shortage of skills in the public sector [9]. Whilst there has been progress in reducing maternal mortality from 160 per 100 000 live births in 2000 to 119 per 100 000 live births in 2017 [10], disparities in the distribution of obstetric care providers, models of care and outcomes remain a problem between the public and private sectors [8]. In South Africa, district hospitals account for 38% of public sector caesarean deliveries with a case fatality rate of 62 per 100,000 caesarean deliveries [8]. Since district hospitals do not have specialist obstetricians or anaesthetists, surgical and anaesthetic skills training and supervision for the medical officers working at this level is critical. However, any PPEs would need to address the challenge of ensuring that patterns of inappropriate care such as the high caesarean delivery rate (74%) of the private sector [11], is not reproduced within the public sector.

Within LMICs, urban and rural districts may require different models of PPEs depending on the available skills and resources in the public and private sectors. In urban areas PPEs could involve private hospital groups and specialist doctors whilst in rural areas the private sector is more likely to consist only of private general practitioners (GPs). For example in India, public-private contracting for obstetric services utilises specialist obstetricians [12]. In some states private obstetricians are contracted in to public facilities while in other states costs for services are subsidized within private facilities [13]. A public-private partnership programme in Gujarat state, India provides private obstetricians with a fixed sum per 100 births

to provide care to eligible women below the poverty line. The remuneration model was specifically chosen as a disincentive to prevent unnecessary caesarean deliveries [14]. Experiences of different models of PPEs for obstetric care in LMICs is limited to specialist obstetricians and poorly documented. We could find no published research of experiences with PPEs for obstetric care in Africa or examples where GPs were contracted.

The objective of this study was to describe the experiences of managers and public sector medical officers, of private GP contracting for caesarean deliveries in five rural district hospitals and one regional hospital in the Western Cape. The study provides insights that could inform current planning for the implementation of national health insurance.

## Methods

### Study design

This exploratory qualitative interview study was part of a mixed-method health systems research study which aimed to describe rural district hospitals' utilization of private GP contracting for caesarean deliveries in the Western Cape province [15].

### Study setting

The setting for this research was the Western Cape province, South Africa. The province has a total population of 7 million [16]. There are 42 district hospitals and 5 regional hospitals in the province. Between April 2021 and March 2022 there were 99630 deliveries in public health facilities in the Western Cape province. Of these, 31% were caesarean deliveries [17].

In South Africa, district hospitals provide level 1 (generalist) services to in-patients and out-patients (ideally on referral from a community health center or clinic). District hospitals have between 30 and 200 beds, a 24-hour emergency service and an operating theatre. Non specialist doctors (medical officers) provide the services together with nursing staff and allied health professionals. Most district hospitals also have community service doctors. These are doctors who have completed a two-year internship and are required to complete a further one year of community service [18]. For obstetric services at district hospital level normal vaginal deliveries are performed by midwives, assisted vaginal deliveries are performed by advanced midwives or medical officers and caesarean deliveries (surgery and anaesthesia) are performed by medical officers.

The public health service is permitted to contract the services of private providers where needed. In the case of obstetric services in the study district, private providers are mainly utilized for theatre services either as a GP surgeon or GP anaesthetist to undertake caesarean deliveries or for emergency gynaecological surgery including ectopic pregnancy, termination of pregnancy and dilatation and curettage following spontaneous miscarriage. They may also be called for an assisted delivery if the establishment medical officer is unable to manage a complicated delivery.

### Sampling

Existing public-private contracting for caesarean delivery services was occurring in the Western Cape province due to public sector human resource shortages in rural district hospitals. Five rural district hospitals within one rural district, and one regional hospital were chosen following engagement with provincial managers and obstetric clinical managers. The five district hospitals had existing contracting arrangements with private GPs at the time of the study, whilst the regional hospital was included in order to explore perceptions of a secondary level hospital obstetrician and manager of potential PPE opportunities.

We utilised purposive sampling to recruit specific participants according to the objectives of the study. At each hospital we obtained a list from the senior medical officer of all GPs with current contracts with the hospital and all were contacted to request participation in an interview. A visit was made to each of the hospitals for in-person interviews. GPs available on the days of the visits and who consented to participate were included to explore their experiences of undertaking work in the public sector. All private GPs agreed to participate but not all were included due to leave, illness or not having time on the day of the visit. At the district hospitals we approached the senior medical officer and one other doctor usually a community service doctor to explore their experiences of the public/ private interface including the reasons for entering into these arrangements. From the district management team the individuals with responsibility for finances, human resources and rural health services were approached and agreed to participate.

## Data collection

Semi-structured interview guides were developed, one for private GPs and one for public providers (medical officers and district managers). In developing the interview guides we drew on a framework previously developed by members of our research team which outlines drivers, challenges and required action for obstetric care in preparing for national health insurance [19]. Types of questions asked to private GPs included: " What has been your experience of working in government hospitals? and 'What are the advantages for you of entering into a contract with a government hospital?"; types of questions asked to district managers and medical officers included: " What services are private GPs contracted to provide?, "Can you tell us what kind of contracts you use for private GPs?, "What experiences have you had of working with private GPs?" and "would it be beneficial for your maternity unit to have greater involvement of private GPs? In what way?". Examples of interview questions organised according to the framework is included as S1 Table.

Interviews were undertaken by three of the investigators (TD, GS and ED) at the hospitals, practices of the private GPs and the district offices. The interviewers were two health economists and a health systems researcher with experience in qualitative interviewing. They had no prior relationships with any of the interviewees. All interviews were conducted in a private consulting room or office and were undertaken in English.

Between April 2021 and March 2022 we undertook a total of 26 semi-structured qualitative interviews (Table 1); including 4 district managers (2 male, 2 female), twelve private GPs (7 male, 5 female) and 8 government employed medical officers (4 male, 4 female), one obstetrician (male) and one hospital manager (male) at the regional hospital. We interviewed all available private GPs contracted to the participating hospitals and all district managers in the specified categories. For interviews with government employed medical officers we continued until no new information was emerging and we determined that data saturation had been reached. Interviews ranged in length from 30 minutes to one hour.

## Data analysis

Interviews were digitally recorded and transcribed by a professional transcriber. Transcripts were validated against the recordings to ensure accurate transcription by two members of the research team (TD and YB). Thematic content analysis using an inductive, iterative approach was applied [20, 21]. Two members of the research team (TD and YB) read transcripts and identified parts of the text dealing with specific issues which were grouped into codes. The full team met to reflect and discuss the emerging codes and to reach agreement on how to sort

**Table 1. Characteristics of hospitals and respondents.**

| | Hospital A | Hospital B | Hospital C | Hospital D | Hospital E | Hospital F |
|---|---|---|---|---|---|---|
| Number of government employed doctors | 0 | 6 | 6 | 10 | 13 | No contracts with private GPs |
| Number of contracted private GPs | One practice consisting of 5 GPs providing a 24 hour service | 2 | 5 | 4 | 1 | |
| Government employed doctors interviewed | 0—no government employed doctors | 1—female | 3 -males | 1-female | 2—female | 1 obstetrician—male |
| | | | | 1—male | | 1 hospital manager—male |
| Private GPs interviewed | 2—both male | 1- male | 2—males | 1-males | 1-male | No contracts with private GPs |
| | | 1- female | 1- female | 3-females | | |
| District management interviews at the district offices | District director—female | | | | | |
| | District human resource manager—male | | | | | |
| | District finance manager—male | | | | | |
| | Manager of rural health services—female | | | | | |

codes into sub-categories, categories and themes. An example of the process followed to develop one thematic area is included in S2 Table.

Efforts to ensure credibility of these research findings included interviewing a range of stakeholders to enable multiple perspectives on the topic from managers, the private GPs being contracted and government medical staff across five different hospital contexts. Trustworthiness was considered by providing a detailed description of the context of rural district hospitals in terms of services provided and human resources.

Major themes are reported in narrative form in this paper. Alongside quotes from interviews we have used descriptors A-E for the district hospitals and F for the regional hospital to protect the anonymity of the hospitals.

## Ethical considerations

Ethics approval was granted by the South African Medical Research Council Human Research Ethics Committee (approval number EC003-2/2021) and permission was also granted by the Western Cape Provincial Department of Health and Wellness (approval number WC_202103_017).

All health professionals who were approached to participate in an interview were read an information sheet outlining the expectations in terms of length of the discussion, the voluntary nature of the participation and measures to ensure confidentiality. No names of interviewees or hospitals have been used in the reporting of the study results to protect confidentiality. Interviewees received refreshments in the form of biscuits to the value of R150 (US$ 14) on completion of the interview as a gesture of thanks for their time. Health professionals who agreed to participate signed an informed consent form.

## Results

Analysis of transcripts revealed six themes related to the experiences of managers and medical officers of public-private contracting arrangements with private GPs. 1) Retaining skilled medical staff who can perform caesarean deliveries in rural district hospitals; 2) Economic advantages of contracting private GPs to rural district hospitals; 3) Flexibility in human resource planning for rural district hospitals; 4) Benefits for private GPs entering into contractual arrangements with public hospitals; 5) Exemplifying a vision of national health insurance and

6) Opportunities to expand PPEs in referral hospitals. These themes are expanded upon in the findings below.

## Retaining skilled medical staff who can perform caesarean deliveries in rural district hospitals

District managers described the justification for contracting with private GPs in their towns. Retaining permanent staff in rural district hospitals is difficult due to high staff turnover and slow recruitment processes. Private GPs were seen to be an important source of surgical and anaesthetic skills. District hospitals have no specialist obstetricians or anaesthetists therefore having medical officers with skills to perform surgery or anaesthesia for caesarean deliveries is critical. Private GPs described their role in on-site training of junior medical officers and community service doctors:

> *If you lose a lot of full-time staff, it can happen quite easily, and it takes about three, sometimes three to four months to fill a post. Especially if you lose your staff that can do caesars (caesarean delivery). Most of the GPs can do Caesars.*

(Private GP, Hospital C)

> *So you need experienced senior doctors at local level and the only senior experts or doctors is your GPs that is there and also to keep them there it is worth your while to also go into a service level agreement with them.*

(District Director)

> *What I also appreciate about him (private GP), when a community service doctor assists, he likes teaching them as well. So that is a good.*

(*Government doctor, Hospital E*)

> *I've got to teach the people here how to do the things here because we work in (name of rural town), you must be able to do a Caesar.*

(Private GP, Hospital C)

Private GPs were also seen as being more permanent in rural areas since they have invested in establishing their private practices. District managers described that maintaining a positive relationship with private GPs is important for the functioning of rural district hospitals:

> *In the rural district we've discovered that you should try and establish a very good working relationship with the skills that's available in a town. You bring in either community service or permanent staff. You can't depend on them. This year you get a community service person that's not experienced or permanent staff come and go. You advertise a post; you appoint the person and the next thing they leave. The private sector that's in that area is more reliable.*

(District finance manager)

> *GP relationship is very important. They're fairly stable in all these towns and they offer skills. So, maintain a good relationship.*

(District human resource manager)

### Economic advantages of contracting private GPs to rural district hospitals

Another consideration in the decision to contract with private GPs was financial. For small rural district hospitals with 50 or fewer total beds, having a full complement of permanent medical officers was perceived to be more costly than having a mix of full-time doctors on the establishment supplemented with private GPs for after-hours cover for caesarean deliveries:

> So, with the budget that we have, it's better to spread it out thinly like this so I worked out that to cover our current contracting requirements, we need about seven doctors and the price of contracting GPs is going to be under two doctors. So I mean from a financial point of view, it makes sense.
>
> (Government doctor, hospital D)

The cost saving was described to be due to avoiding the additional cost of leave, overtime pay and pension which permanent staff receive but not contracted private GPs as these participants explained:

> For the smaller hospitals to cover a twenty four hour, seven day shift with doctors, you need at least seven to nine doctors. If you then bill for permanent doctors, then you need to think about their leave, there's maternity leave. So for a small hospital, it's not economical to employ seven full-time, I mean, we're working on one point six million rand for a doctor with all the benefits.
>
> (District finance manager)

> If you want to run this hospital with a full-time component, you'll have to get about ten permanent posts. You must make provision for the maternity leaves, the sick leaves, the annual leave. If you get the right balance between GPs and full-time it's financially I think better. You must consider a number of full-time people to render the service, you know, the after-hour calls and things like that. So you must have a good balance. Then it's more economical, I think.
>
> (Private GP, Hospital C)

There was acknowledgement from public sector medical officers and a few private GPs that the rates paid were suboptimal given the need for the GPs to be available on-standby and the payment being different for standby versus on site, as these two doctors explained:

> You know it costs them maybe six hundred and fifty rand per hour but the doctor only gets four fifty. So it's not just about the money but it will go a long way if there's a financial incentive as well. Because remember I need to be available. I can't go anywhere.
>
> (Private GP, Hospital D)

> If you're doing a whole weekend of standby and you're getting paid as a locum, you actually have to be at home and you can't move anywhere for that whole weekend. You need to be in a certain radius and you're only getting paid a third unless you get called out and then you get additional, say three hours for the procedure, but you essentially give up a whole weekend and you're only getting a third of that hours.
>
> (Government doctor, Hospital D)

### Flexibility in human resource planning for rural district hospitals

District managers shared their experiences staffing rural hospitals and the need for context specific, flexible planning that allows for a mix of public and private doctors:

*There's no way that you can have a pie in the sky approach and say I'd like to have seven doctors to cover a thirty bed hospital. So you're going to have to be innovative and use different contracts. I don't know whether our public service contracts must become more innovative from public service and admin national side. They need to give us a more flexible kind of contract.*

(District human resource manager)

*We decided at that stage that a mixed model of permanent staff, community service staff, and the GPs in town. So you're left with trying to balance and not be too reliable on the private sector. And that is now different from town to town. We decided you're going to have to look at the local reality. What doctors are available in private, are they skilled and use them, and then supplement them with community service and permanent staff.*

(District finance manager)

The district director and senior medical officer in one of the hospitals cautioned that the staffing balance should not shift towards having a majority private GPs at the district hospitals because they feared that this would compromise clinical governance and adherence to public sector protocols as these participants described:

*this whole thing about the balance between private GPs and full-time MOs. Three years ago there were only four of them running a seventy five bed hospital which is extremely busy, offering caesars, whatever else, but then they had like ten private GPs that were employed on contracts doing various things and doing after hours calls. I think this balance between the two, that to me is out of balance. And then you run into all sorts of problems with clinical governance. They're not all good you know, some of them are excellent, and some of them aren't.*

(Government doctor, hospital D)

*I don't have a preference for state doctors. But as you know, you can train them in our core package of service because if you are getting people for contracts over weekends in particular, you know, they are not always used to our protocols.*

(District director)

In terms of disadvantages of these arrangements, private GPs contracted to two of the district hospitals voiced that they would prefer more job security since their contracts with the public sector are renewable every three years and they don't receive any of the benefits that permanent public sector doctors receive, as these GPs explained:

*So we are applying for our job every three years now but we don't have any other advantages like permanent people.*

(Private GP, Hospital C)

*The one thing that I don't like is the fact that we've got three-year contracts. I don't think that's fair of the state to give us three year contracts and then you sit at the end of the three years with all this insecurity and worry are you going to get it again? And they say no, it's just*

*a formality and you'll get it again. I think what they should do is after every three years, let's write an exam or let's just do an interview as far as whether you're still on top of your game. But we need more security as far as job security is concerned. You know, we don't get pension, we don't get a holiday, we don't get anything. I feel from government side, they should say, look, you guys make sure that there's security as far as service delivery is concerned. We're going to give you a ten-year contract or five year contract at least.*

(Private GP, Hospital B)

## Benefits for private GPs entering into contractual arrangements with public hospitals

Interviews with private GPs revealed several advantages to them of having a contract to work in the public district hospital. These included staying up to date with clinical protocols and maintaining surgical and anaesthetic skills. None of the private GPs provided obstetric care in their private practices because of the high cost of indemnity insurance for obstetric care. Therefore, their only way to maintain skills in obstetric surgery and anaesthesia was through engaging in state work:

*In the private practice you lose contact with those protocols in the newest ways to do things. So it's nice having them (hospital doctors) here and ask, I've got a pre eclampsia now. What do you want to do? Give me the protocol. It's difficult to keep up with all the new protocols.*

(Private GP, Hospital C)

*The reason why I'm both in the state and private practice is because it helps to have the flexibility of the private practice but you still don't lose your skills like your surgical skills and everything like the C-section skills that you can only do in the hospital because obviously none of us are taking out that, insurance.*

(Private GP, Hospital C)

Private GPs also spoke of the benefit of learning from visiting specialists from regional hospitals who do monthly outreach to district hospitals:

*It gives me personally a lot of job satisfaction and lot of contact, with good medicine. With specialists.*

(Private GP, Hospital B)

*We get proper input from visiting specialists on a monthly basis. We do a round with them and you get their input and you see what's the latest to look out for, what's the latest to implement this and I couldn't understand how you don't want to stay ahead and be part of that. I don't know. It didn't make sense to me. I mean we're lucky to have that come to us. We sit in (name of rural town), we can't go to lectures every week and go to seminars. But these people come to us, they see our patient, we're there.*

(Private GP, Hospital B)

Having a contract with the district hospital also provided GPs with a source of income that was more reliable in a rural setting than their private income as these GPs described:

*Here, there is not as much private healthcare as you would find in Cape Town, where you can isolate yourself necessarily from what's happening, in the public sector. So, that gives us a chance to do a bit of both and in that way helps supplement your income. So, your income isn't reliant on your state income and it's not reliant on your private income, but the combination together makes up your, total package.*

(Private GP, Hospital A)

*You kind of need the capital from both sides to make ends meet.*

(Private GP, Hospital C)

## Exemplifying a vision of national health insurance

District managers, government employed doctors and private GPs described the unique contribution and value of public-private contracting arrangements for rural towns as the insured and uninsured population receive care from the same doctors who work in both sectors. This was described to represent the vision of national health insurance as these respondents explained:

*I think, you know, it's nice when you can do both and treat them the same. When you have someone that treats a patient irrespective of their financial situation equally, I feel like it's as close to the NHI, the vision that NHI has, as we can get. You know it's irrespective of your financial background. You're gonna get the same doctor. You're going to get the same hospital. You're going to get the same nursing staff. So it's nice having that, so that the patients don't feel forced to go to a private doctor because it's the same doctor in the state and the private setting. So you can go to Dr. X's practice but if you come to the hospital, Dr. X will see you tomorrow morning anyway.*

(Government doctor, Hospital B)

*This is the collegial cordiality that they also have here in this district, you know, to help each other, irrespective if you are on this government's establishment or you are a private GP. We are here to focus on the patient and to see where we can do the best for our patient.*

(District director)

Only one of these rural towns had a private hospital therefore in the event of an emergency private GPs had to admit their insured patients to the public hospital. This led to them having a vested interest in the functioning of the hospital and wanting to instil a sense of local pride in the health system, as these GPs described:

*I want (name of rural town) people to work in this hospital because if they walk in the street and the hospital is dirty, they feel ashamed about the hospital is dirty. If you have permanent people, they're permanent but they're not permanent. They go in two year's time. Their kids need to go to school in Cape Town or whatever. Then they're off. Off they go. I want people that live here, that know all the people in the town, that have vested sort of interest in this town and into the hospital and to health and general health in the hospital. And that I don't think you get with sole state practitioners.*

(Private GP, hospital B)

*I think it's absolutely essential that we remain in state work for many reasons. I think we've got lots of experience first of all. We, as people that benefit from the community and from the town and that live here forever, we should stay in the system and make it work at all cost, and build some pride into your system.*

(Private GP, hospital B)

## Opportunities to expand PPEs in referral hospitals

Respondents from the regional hospital spoke of the need for alternative models of public-private contracting. These participants described how the shift in population affordability of private care has led to greater demands on the regional hospital. A formal arrangement between the public and private sectors was deemed to be necessary for continuity of patient care given the movement of patients between sectors that currently function completely separately:

*You'll find what was happening quite a lot is a client will go to a private GP for seven months of her pregnancy, then her medical insurance is finished. Now she comes across to us or she goes for the antenatal period to a GP and comes to us to deliver. And she arrives with us basically unbooked because there's no connection between the two. So at least to get that connection, so that when they do land in our services, it's kind of in line with what we expect.*

(Manager rural health services)

At the regional hospital level where specialist obstetricians and anaesthetists exist, their need was not for contracting-in of private GPs but rather for contracting out services to specialists in private hospitals, especially planned elective procedures, due to high patient load and insufficient operating theatre resources:

*So because the obesity rate has increased, we are finding that more district hospitals need to refer all these patients to us. So we are doing much more caesarean sections both because of failed induction of labour and because they are obese and had a previous caesarean section, it cannot be safely done at a district hospital. So, our numbers are increasing with these kinds of patients, because these procedures are planned, we can then employ our private partners, it's more planned procedures that we can actually plan with them.*

(Obstetrician, hospital F)

A government doctor from hospital E which was in the only town with a private hospital, described how her hospital could benefit from being able to outsource services such as ultrasounds to the private hospital:

*If you think about how many ultrasounds we do ourselves If we can establish a relationship with next door (name of private hospital), please can you do twenty, gynae ultrasounds for us? We manage our antenatal or that relationship, I think it would be fantastic really.*

(*Government doctor, hospital E*)

The regional hospital manager also spoke of the advantage of being able to contract a local private hospital instead of sending a patient to a city for care which required costly transport:

*In rural areas, I think the concept of being able to offer patients local care working with private providers is really important.*

(Hospital manager, hospital F)

## Discussion

This research described experiences of public-private contracting for obstetric care within rural district hospital settings in South Africa. Qualitative interviews with medical officers, private GPs and district managers revealed justifications for entering into these partnerships, including retention of doctors with anaesthetic and surgical skills and economic considerations. The arrangements held benefits for both the public sector and the contracted private GPs and was deemed to be an example of how the two sectors can work together to operationalise the vision of national health insurance. Perspectives of a specialist and manager from a regional hospital provided insight into the need for different public-private solutions for this level of care in which contracting out of services should be considered.

There is a paucity of literature on PPEs for health care from Africa. A systematic review that included 52 PPE initiatives in Southern Africa found none that included contracting-in of private providers and none that were related to the provision of obstetric care [2]. A 2017 Cochrane review of contracting-out clinical health services in LMICs found only two studies neither of which were from Africa [22]. An evaluation of a prominent PPE in Lesotho to build and operate a tertiary hospital in Maseru, contracted to a South African-based private hospital group, found that the costs to government were greater than was forecast and the Ministry of Health had limited capacity to plan, procure, pay for and manage the contract [3]. Domestic capacity to manage complex contracts is an important consideration in PPEs in LMICs.

Provision of hospital services including essential surgery and anaesthesia in rural districts is an important but neglected component of achieving UHC [23]. As described by participants in our study, there are inherent challenges for public health systems to adequately staff low volume rural district hospitals as the number of cases requiring surgery may be too low to warrant permanent on-site doctors with surgical skills to cover 24 hours, 7 days a week. Furthermore, a fully public sector staffed doctor complement for small rural hospitals may not make financial sense due to low case load. Entering into local public-private arrangements is thus an important option for consideration where providers can be paid for a specific number of hours without the added expense of salary benefits such as leave and pension. A literature review exploring staffing of remote, rural areas in LMICs, identified partnerships with the private sector as one policy option for such contexts where low patient numbers may warrant a mix of private and public sector employed staff [24].

In most LMICs, including South Africa, medical education and training programs which remain urban tertiary hospital focused are not adequately preparing junior doctors for rural and remote practice where skills in essential surgery and anaesthesia are required. The sustainability of any GP contracting arrangement, such as described in this paper, will require ensuring that medical education programmes include basic surgical and anaesthetic skills training so that GPs opening practice in rural areas have the required skills to provide these services for district hospitals where needed [25].

Our study has revealed that there are benefits to both the public sector and private GPs entering into contractual relationships. The public sector benefits from the added skilled human resources to fill staffing gaps, surgical skills training and financial savings whilst the private GPs are able to maintain surgical skills, receive ongoing medical education from visiting specialists and subsidise their private earnings. The viability of rural GP practices are

under threat in many countries yet these practitioners play a vital role in meeting the health needs of rural communities [26]. In South Africa, the private insured population in rural areas is lower than in cities. Coverage of private medical insurance was 15% of the South African population in 2020, but this ranged from 25% of individuals in the Western Cape to 8% of individuals in the most rural provinces of Mpumalanga and Limpopo [5]. This situation limits the earning potential of private practitioners in rural areas; however, in the absence of these practices patients would either need to seek care from the public sector or travel to a city.

This study has identified that different models of PPEs may be needed for different levels of care. Whilst contracting in skills was beneficial to rural district hospitals, the needs of a large regional hospital were described to require a different approach, namely out-sourcing services to private specialists and hospital groups. Ultimately the choice of PPE model for a given health system context should be one that expands and improves the quality of services for defined populations at an affordable and economically efficient cost [4].

## Limitations

Only one province, the Western Cape was included. Experiences and needs may be different in other parts of the country. Private obstetric specialists were not interviewed. Their perspectives are important. Transferability of these findings should be limited to similar rural district hospital settings.

Of note, few disadvantages or challenges with these contracting arrangements were voiced. The interviewers had no prior interactions or relationships with any of the interviewees nor any pre-conceived biases about benefits or pitfalls associated with these contracting arrangements. Reflexive discussions amongst the research team were held throughout the data collection process to reflect on emerging findings and to amend the interview guide to ensure that all aspects of the topic were explored.

## Conclusion

Many countries in sub-Saharan Africa are required to confront the need to shape their health systems to include private systems in the achievement of public goals but it is essential that these solutions are tailored to local contexts. Leveraging private sector expertise and capacity to support the public health system has the potential to contribute to universal health coverage irrespective of the purchasing power of citizens.

## Supporting information

**S1 Table. Interview guide questions organised according to the framework [19] of drivers, challenges and required action for obstetric care in preparing for national health insurance.**
(DOCX)

**S2 Table. Example of the process of data analysis to develop codes, sub-categories, categories and themes.**
(DOCX)

**S1 Checklist. Standards for Reporting Qualitative Research (SRQR) checklist.**
(PDF)

## Author Contributions

**Conceptualization:** Tanya Doherty, Sue Fawcus, Geetesh Solanki.

**Data curation:** Tanya Doherty, Yvandi Bartmann.

**Formal analysis:** Tanya Doherty, Emmanuelle Daviaud, Yvandi Bartmann, Geetesh Solanki.

**Funding acquisition:** Tanya Doherty, Sue Fawcus.

**Methodology:** Sue Fawcus.

**Project administration:** Tanya Doherty.

**Supervision:** Sue Fawcus.

**Writing – original draft:** Tanya Doherty.

**Writing – review & editing:** Sue Fawcus, Emmanuelle Daviaud, Yvandi Bartmann, Geetesh Solanki.

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
