## [Decision Letter · Decision Letter 0]

4 Nov 2022

PGPH-D-22-01557

Perspectives of public-private contracting for caesarean delivery in rural district public hospitals: A qualitative interview study

Dear Dr. Doherty,

Thank you for submitting your manuscript to PLOS Global Public Health. After careful consideration, we feel that it has merit but does not fully meet PLOS Global Public Health’s publication criteria as it currently stands. Therefore, we invite you to submit a revised version of the manuscript that addresses the points raised during the review process.

We look forward to receiving your revised manuscript.

Kind regards,

Patrick A. Palmieri, DHSc, DPhil(Hon), EdS, MBA, MSN, PGDip(Oxon), ACNP, RN, CPHRM, CPHQ, FFNMRCSI, FAAN

Academic Editor

Journal Requirements:

2. Since your data is not available for proprietary reasons, please explain via email why the data is not available. Please also include the contact information for the third party organization that should be contacted should other researchers want to request access to this data and please include the full citation of where the data can be found. We also request that you verify with us via email that any researcher will be able to obtain the data set in the same manner that the you have obtained it. If you feel you are unwilling or unable to adhere to this policy, please explain your reasons by return email and your exemption request will be escalated to the editor for approval. Your exemption request will be handled independently and will not hold up the peer review process, but will need to be resolved should your manuscript be accepted for publication. One of the Editorial team will be in touch if they require more information.

Additional Editor Comments (if provided):

There are multiple observations to address limitations in reporting the study and to improve the clarity of presentation. When the authors undertake the revisions, the SRQR (Standards for reporting qualitative research) checklist should be applied to the manuscript to make sure all the recommended reporting elements have been properly addressed (https://www.equator-network.org/reporting-guidelines/srqr/). The completed SRQR checklist needs to be returned with the revised manuscript with the page and line or paragraph numbers indicated for each reporting element. Please make sure to review the supporting document for the checklist to make sure ALL the criteria for each element are properly addressed.

Reviewers' comments:

Reviewer's Responses to Questions

**Comments to the Author**

1. Does this manuscript meet PLOS Global Public Health’s publication criteria? Is the manuscript technically sound, and do the data support the conclusions? The manuscript must describe methodologically and ethically rigorous research with conclusions that are appropriately drawn based on the data presented.

Reviewer #1: Yes

Reviewer #2: Partly

2. Has the statistical analysis been performed appropriately and rigorously?

Reviewer #1: Yes

Reviewer #2: N/A

3. Have the authors made all data underlying the findings in their manuscript fully available (please refer to the Data Availability Statement at the start of the manuscript PDF file)?

Reviewer #1: Yes

Reviewer #2: Yes

4. Is the manuscript presented in an intelligible fashion and written in standard English?

Reviewer #1: Yes

Reviewer #2: Yes

5. Review Comments to the Author

Reviewer #1: To the bet of my knowledge, the manuscript meets PLOS Global Public Health's publication criteria.

All data pertaining to the manuscript has been provided except for transcripts which the authors retained for participant privacy.

The manuscript is well written, however, the authors should format paragraphs by justifying so that the text is distributed evenly within the margins.

Under the methods section, I advise authors to format subtitles by making them bold.

To have a better read, more punctuation should be done by the authors, especially the use of commas.

Reviewer #2: This article explores the challenge faced by many LMICs in the pursuit of health care service provision for the population. The authors explore the experiences and perceptions of providers in private and public health care facilities to public-private contracting for caesarean delivery in rural district public hospitals in the Western Cape Province of South Africa.

There are several recommended areas of improvement

It would have been useful to have line numbers and page numbers in the manuscript

Introduction

• Please include statistics on the uninsured population that use the private sector and the insured population that use the public sector services

• Is there any available financial data on the costs of public private contracting in South Africa

• The authors indicate there has been a decline in maternal mortality. It would be useful to indicate during what time frame this has occurred and the earlier material mortality rates.

• It is unclear if the high caesarean delivery rate in the private sector is considered a pattern of inappropriate care (this paragraph last two sentences)

• Can the different models of PPEs documented in LMICs be described in greater detail, perhaps with specific examples from other countries

Under Methodology

• What is the population of this province?

• How many deliveries occur in the private facilities and how many in the public facilities?

• How many caesarean deliveries take place in the private facilities and how many in the public facilities in the province?

• How many district hospitals and regional hospitals were in this province? How many regional hospitals are in this province?

• Were these five district hospitals selected because they were the only ones that had existing contracting arrangements?

• Were the hospitals labelled A, B, C, D, E, and F? Which ones were the district hospitals, and which one is the Regional Hospital?

• What language was used in the interview? Was there any translation of the transcripts required?

• Was any prior literature used to develop the interview questions?

• What types of questions were included in the interview schedule?

• What were the selection criteria for participants?

• What were the roles of the 3 district managers mentioned under data collection? In the results section quotes are attributed to District Director, District Human Resource Manager, District Finance Manager (hospital affiliation not provided) , Medical Manager, Manager Rural Health Services and Hospital Manager. Please clarify.

• Where was the interview conducted at the facilities? Was it a private room?

• Interview duration (range of time and average length of time)?

• What measures were taken to ensure confidentiality? Did the interviewees receive an honorarium for their participation?

• Who transcribed the interviews (a team member or a RA)?

Results

• A small table which breakdowns the gender, role and facility would be useful for the reader

• Outline in the first paragraph the six thematic areas that would be covered in the results section. Not all are mentioned

• It is unclear if all of the interviewees shared similar opinions and if both subcategories of doctors are included, eg “Public sector hospital manager and doctors described the justification for entering into contracting agreements with private GPs in their towns” Does doctors refer to both private sector and government employed doctors? Did all of the doctors interviewed justify contracting agreements with private GPs in their towns? There were no quotes from government employed doctors under the theme.

• Clarify which quotes emerged from personnel at district hospitals or the regional hospitals

• Please provide numbers alongside some, several to clarify for the reader how many of the interviewees shared this view

• There is an observably higher representation of quotes from private GPs and managers compared to government employed doctors in the results section. Recommendation to include quotes from all categories of interviewees

o 4 quotes from Government employed doctors – medical officer Hospital D, medical officer Hospital B, Obstetrician from Hospital F (two quotes attributed)

o 14 quotes from Private GPs – 7 from private GP at hospital C, 5 from private GP at hospital B, 1 from private GP at hospital A and one from private GP at hospital D

o 12 quotes from manager level

Discussion

This section would be improved by comparing the results with the perceptions of and models of PPEs utilized in other LMICs, or other provinces in South Africa. This can include the financial impact of utilizing PPEs on government’s health care expenditures, levels of efficiency, public acceptance and health statistics. Please provide statistics and references where needed.

In South Africa, the private insured population in rural communities is lower than in cities which limits the earning potential of private practice (statistics and citation).

6. PLOS authors have the option to publish the peer review history of their article (what does this mean?). If published, this will include your full peer review and any attached files.

**Do you want your identity to be public for this peer review?** For information about this choice, including consent withdrawal, please see our Privacy Policy.

Reviewer #1: No

Reviewer #2: No

---

## [Decision Letter · Decision Letter 1]

16 Feb 2023

PGPH-D-22-01557R1

Perspectives of public-private contracting for caesarean delivery in rural district public hospitals: A qualitative interview study

Dear Dr. Doherty,

Thank you for submitting your manuscript to PLOS Global Public Health. After careful consideration, we feel that it has merit but does not fully meet PLOS Global Public Health’s publication criteria as it currently stands. Therefore, we invite you to submit a revised version of the manuscript that addresses the points raised during the review process.

I would like to sincerely apologise for the delay you have incurred with your submission. We have now received three completed reviews; the comments are available below. The two reviewers that previously provided comments to this work are happy with the revisions, however the additional reviewer has raised significant scientific concerns about the study that need to be addressed in a revision. 

Please revise the manuscript to address all the reviewer's comments in a point-by-point response in order to ensure it is meeting the journal's publication criteria. Please note that the revised manuscript will need to undergo further review, we thus cannot at this point anticipate the outcome of the evaluation process.

We look forward to receiving your revised manuscript.

Kind regards,

Miquel Vall-llosera Camps

Staff Editor

Journal Requirements:

Additional Editor Comments (if provided):

Reviewers' comments:

Reviewer's Responses to Questions

**Comments to the Author**

1. If the authors have adequately addressed your comments raised in a previous round of review and you feel that this manuscript is now acceptable for publication, you may indicate that here to bypass the “Comments to the Author” section, enter your conflict of interest statement in the “Confidential to Editor” section, and submit your "Accept" recommendation.

Reviewer #1: All comments have been addressed

Reviewer #2: All comments have been addressed

Reviewer #3: (No Response)

2. Does this manuscript meet PLOS Global Public Health’s publication criteria? Is the manuscript technically sound, and do the data support the conclusions? The manuscript must describe methodologically and ethically rigorous research with conclusions that are appropriately drawn based on the data presented.

Reviewer #1: Yes

Reviewer #2: Yes

Reviewer #3: Partly

3. Has the statistical analysis been performed appropriately and rigorously?

Reviewer #1: (No Response)

Reviewer #2: N/A

Reviewer #3: N/A

4. Have the authors made all data underlying the findings in their manuscript fully available (please refer to the Data Availability Statement at the start of the manuscript PDF file)?

Reviewer #1: (No Response)

Reviewer #2: No

Reviewer #3: Yes

5. Is the manuscript presented in an intelligible fashion and written in standard English?

Reviewer #1: (No Response)

Reviewer #2: Yes

Reviewer #3: Yes

6. Review Comments to the Author

Reviewer #1: (No Response)

Reviewer #2: Thank you to the authorship team for considering the feedback when revising your manuscript. It is a well written piece that adds to the literature in this area. Some minor corrections in grammar are recommended, having said that, the piece flows easily and all queries have been answered. As no statistical analysis was done, I have indicated N/A with regards to statistical analysis.

Reviewer #3: Thank you for the opportunity to review the article "Perspectives of public-private contracting for caesarean delivery in rural district public hospitals: A qualitative interview study" submitted to PLOS Global Public Health. The majority of the feedback for this review is provided in the marked version of the manuscript pdf to assist the authors with the recommended revisions. The introduction can be improved with editing and minor revisions, possibly some rearranging, as recommended in the marked version of the manuscript. Also, there seems to be some information in the methods section that is more specific to the background than to the methods. The methods section requires multiple revisions to comply with the reporting elements of the SRQR which was attached as the reporting guideline. For example, the information about trustworthiness is largely not reported in the methods section. More importantly, the data analysis area requires improvement as the process outlined by Graneheim & Lundman is largely not reflected in the current description of the study. Similarly, the SRQR addresses the coding tree, an important feature in the process described by Graneheim & Lundman (see figures 1 and 2). The discussion section requires more information about the similarities and differences in the findings of this study with other studies, from LMICs as well as HICs. The current discussion is mechanical in summarizing the findings with little insightful comparisons from the larger literature. As a notation, the "paucity of literature" evidenced by the authors with the Cochrane review is incorrect due to the inclusion criteria. In this regard, the current study would not be included in the cited Cochrane review. Finally, the manuscript lacks clarity in some areas due to the wordy, and often long, sentences. For this reason, the manuscript requires a thorough editing to produce clear and concise sentences. Multiple examples are provided in the attached pdf. Please keep in mind the feedback in the attached pdf is quite critical, but in a constructive manner to address the observations, limitations, and recommendations. Once the revisions are completed, the article should be a good contribution to the literature.

7. PLOS authors have the option to publish the peer review history of their article (what does this mean?). If published, this will include your full peer review and any attached files.

**Do you want your identity to be public for this peer review?** For information about this choice, including consent withdrawal, please see our Privacy Policy.

Reviewer #1: No

Reviewer #2: No

Reviewer #3: No

---

## [Editor Report · Decision Letter 2]

10 Apr 2023

Perspectives of public-private contracting for caesarean delivery in rural district public hospitals: A qualitative interview study

PGPH-D-22-01557R2

Dear Professor Doherty,

We are pleased to inform you that your manuscript 'Perspectives of public-private contracting for caesarean delivery in rural district public hospitals: A qualitative interview study' has been provisionally accepted for publication in PLOS Global Public Health.

Best regards,

Julia Robinson

Executive Editor